# The Best of Both Worlds: Bridging Quality and Diversity in Data Selection with Bipartite Graph

Minghao Wu [1]  Thuy-Trang Vu [1]  Lizhen Qu [1]  Gholamreza Haffari [1]

## Abstract

The performance of large language models (LLMs) is strongly influenced by the quality and diversity of data used during supervised fine-tuning (SFT). However, current data selection methods often prioritize one aspect over the other, resulting in suboptimal training outcomes. To address this, we formulate data selection as a set cover problem and present GRAPHFILTER, a novel approach that balances both quality and diversity in data selection. GRAPHFILTER models the dataset as a bipartite graph connecting sentences to their constituent n-grams, then employs a priority function that combines quality and diversity metrics multiplicatively. GRAPHFILTER iteratively selects sentences with the highest priority, removes covered n-grams from the bipartite graph, and recomputes priorities to reflect the changing data landscape. We validate GRAPHFILTER using three model backbones across six widely-used benchmarks, demonstrating that it outperforms nine existing baselines in both model performance and computational efficiency. Further analysis shows that our design choices lead to more effective subset selection, underscores the value of instruction diversity, and provides insights into how quality and diversity interact with different subset sizes.

## 1. Introduction

Large language models (LLMs) have significantly advanced the field of natural language processing (NLP), enabling models to generate coherent and contextually relevant text across a variety of tasks (Ouyang et al., 2022; Sanh et al., 2022; OpenAI, 2023; Touvron et al., 2023a;b; Anil et al., 2023; Mesnard et al., 2024; Yang et al., 2024). Central to the success of these models is the quality and diversity of the data used during supervised fine-tuning (SFT). Fine-tuning on high-quality data ensures that the model learns accurate language patterns and responds appropriately to inputs (Wang et al., 2023; Zhou et al., 2023), while diversity in the data allows the model to generalize across different contexts and topics (Abbas et al., 2023; Maharana et al., 2024). However, the vastness of available SFT data presents a challenge: selecting a subset of data that balances both quality and diversity to optimize model performance.

Recent methods for data selection often prioritize either quality or diversity, rarely achieving an optimal balance of both. Approaches that focus exclusively on quality may overlook the variety of language patterns necessary for effective generalization (Marion et al., 2023; Ankner et al., 2024). Conversely, methods emphasizing diversity might include lower-quality data, which could negatively impact model performance (Abbas et al., 2023; Lu et al., 2024). This focus can result in models that either overfit to specific data patterns or underperform due to the inclusion of irrelevant or poor-quality data. Hence, it is crucial to develop a data selection strategy that simultaneously maximizes both data quality and diversity for effective supervised fine-tuning.

In response to this challenge, we formulate data selection as a *set cover problem* and propose a novel method, GRAPHFILTER, which models both *diversity* and *quality* in data selection. The set cover problem aims to select the smallest collection of subsets to cover every element in a given universal set (Garey & Johnson, 1979). To achieve this in data selection, GRAPHFILTER models the dataset as a bipartite graph, where sentences and n-grams are represented as two distinct sets of nodes, with edges indicating the presence of n-grams in sentences. This bipartite structure allows us to prioritize sentences that introduce unique n-grams, thereby maximizing the diversity of the selected subset. In addition to diversity, GRAPHFILTER incorporates quality into the selection process by re-ranking sentences based on a quality metric. To balance these two aspects, we employ a priority function that combines quality and diversity metrics multiplicatively. Concretely, we use SUPERFILTER (Li et al., 2024a) as the quality metric. It measures the infor-

[1]Department of Data Science & AI, Monash University, Melbourne, Australia. Correspondence to: Minghao Wu <minghao.wu@monash.edu>.

*Proceedings of the 42nd International Conference on Machine Learning*, Vancouver, Canada. PMLR 267, 2025. Copyright 2025 by the author(s).

mativeness of a response by comparing its perplexity when conditioned on the instruction with its standalone perplexity. Moreover, we leverage Term Frequency-Inverse Document Frequency (TF-IDF) scores for n-grams within sentences as the diversity metric. The priority function multiplies these two measures, assigning higher priority to sentences that are both informative (high-quality) and contribute substantially to n-gram diversity. During selection, GRAPHFILTER iteratively chooses the sentence with the highest priority, updates the bipartite graph by removing covered n-grams, and recalculates priorities based on the updated graph. By balancing diversity and quality in this manner, GRAPHFILTER effectively build a subset of examples that is both high-quality and broadly representative of the entire dataset.

To demonstrate the effectiveness of GRAPHFILTER, we conducted extensive experiments, comparing GRAPHFILTER against nine baseline approaches using three model backbones across six widely-used benchmarks. Our empirical results indicate that GRAPHFILTER significantly outperforms recent state-of-the-art baselines and achieves notably better computational efficiency. Specifically, in terms of overall performance, GRAPHFILTER substantially outperforms all the baseline approaches across three model backbones, and is significantly more efficient than most baselines without requiring GPUs for computation. GRAPHFILTER outperforms the baselines by up to $+2.37$ for GEMMA-2-2B, $+3.02$ for MISTRAL-7B-V0.3, and $+3.38$ for LLAMA-3-8B, and is significantly more efficient than these baselines without requiring GPUs for computation. Furthermore, we perform an in-depth analysis to validate the effectiveness of our design choices in GRAPHFILTER, examine the characteristics of the selected subsets and the importance of instruction diversity, and investigate the impacts of quality and diversity under various data scales.

In summary, the contributions of this work are threefold:

- We frame data selection as a set cover problem and introduce a novel approach, GRAPHFILTER, which leverages a bipartite graph structure to balance both diversity and quality. This bipartite graph connects sentences to their constituent n-grams, enabling the construction of a subset of examples that is not only high-quality but also broadly representative of the entire dataset (see Section 3).
- Through experiments using three model backbones across six widely-used benchmarks, we demonstrate that our method, GRAPHFILTER, surpasses existing data selection strategies, achieving significantly better computational efficiency (see Section 4).
- Our detailed analyses provide valuable insights into the design choices of GRAPHFILTER, the characteristics of the selected subset, and the importance of quality and diversity in relation to the subset sizes (see Section 5).

## 2. Related Work

**Data Engineering for Large Language Models** The success of recent large language models (LLMs) largely relies on the data used during their training process (Zha et al., 2023). State-of-the-art LLMs are generally trained on vast corpora (OpenAI, 2023; Team et al., 2024; Dubey et al., 2024). A significant area of research focuses on curating high-quality corpora for pre-training these models (Raffel et al., 2020; Computer, 2023; Soldaini et al., 2024; Penedo et al., 2024). Furthermore, Wang et al. (2023) demonstrate that LLMs are capable of synthesizing high-quality datasets for supervised fine-tuning, which leads to a surge of research on dataset synthesis (Xu et al., 2023; Li et al., 2023; Gunasekar et al., 2023; Ding et al., 2023; Cui et al., 2023; Wu et al., 2024; Chen et al., 2024a; Xu et al., 2024). These research efforts facilitate the synthesis of large-scale datasets containing billions of tokens for various purposes, resulting in a significant demand for selecting valuable subsets.

**Data Selection** Data selection strategies aim to identify the most informative data subsets for training or fine-tuning models by considering quality and diversity. Quality-focused approaches prioritize metrics like complexity, difficulty, or informativeness (Marion et al., 2023; Chen et al., 2024b; Liu et al., 2024; Li et al., 2024b;a), but may neglect the range of language patterns needed for generalization. Conversely, diversity-focused methods capture a broad spectrum of linguistic patterns and contexts, potentially incorporating lower-quality data that could impair model performance (Abbas et al., 2023; Lu et al., 2024).

**Ours** To overcome limitations in current data selection methods, we propose GRAPHFILTER, a novel approach that represents the dataset as a bipartite graph of sentences and their n-grams. By balancing quality and diversity with a priority function, our method improves model performance across various downstream tasks.

## 3. Methodology

In this section, we first introduce the data selection problem for supervised fine-tuning in Section 3.1. Subsequently, we describe the modeling of the dataset as a bipartite graph in Section 3.2. Finally, we explain the re-ranking of the graph nodes using a priority function that integrates quality and diversity metrics in data selection in Section 3.3.

### 3.1. Data Selection Problem

The data selection problem involves the challenge of identifying and selecting the most relevant and informative subset of supervised instances from a larger dataset to fine-tune large language models (LLMs). Formally, let $\mathcal{D} = \{(x_i, y_i)\}_{i=1}^{N}$ be the supervised fine-tuning (SFT)

---

**Algorithm 1** GRAPHFILTER

---

1: **Input:** $\mathcal{U} = \{u_i\}_{i=1}^N$, the set of sentence nodes; $\mathcal{V} = \{v_j\}_{j=1}^M$, the set of n-gram nodes; $\mathcal{E} \subseteq \mathcal{U} \times \mathcal{V}$, the set of edges between sentence nodes and n-gram nodes; $k$, the data selection budget; $\phi(u)$, the priority function for each $u \in \mathcal{U}$;
2: **Output:** The selected subset $\mathcal{S}$;
3: $\mathcal{S} = \emptyset$
4: **while** $|\mathcal{S}| < k \wedge \mathcal{U} \neq \emptyset$ **do**
5:     $u^* \leftarrow \arg\max_{u \in \mathcal{U}} \phi(u)$    {Select the sentence with the highest priority}
6:     $\mathcal{V}_{u^*} \leftarrow \{v \in \mathcal{V} \mid (u^*, v) \in \mathcal{E}\}$ {Find n-gram nodes connected to $u^*$}
7:     $\mathcal{S} \leftarrow \mathcal{S} \cup \{u^*\}$ {Add $u^*$ to the selected set}
8:     $\mathcal{U} \leftarrow \mathcal{U} \setminus \{u^*\}$ {Remove $u^*$ from the remaining sentences}
9:     $\mathcal{E} \leftarrow \mathcal{E} \setminus \{(u^*, v) \mid v \in \mathcal{V}_{u^*}\}$ {Remove edges connected to $u^*$}
10:     **for all** $v \in \mathcal{V}_{u^*}$ **do**
11:       $\mathcal{E} \leftarrow \mathcal{E} \setminus \{(u, v) \mid u \in \mathcal{U}\}$ {Remove edges connecting to $v$}
12:     **end for**
13: **end while**

---

dataset, where $x_i$ represents the instruction and $y_i$ its corresponding response for the $i$-th training instance. Our aim is to select a subset $\mathcal{S}_\pi$ of size $k$ from $\mathcal{D}$, utilizing the data selection strategy $\pi$, where $k$ is the *data selection budget*. The objective is to determine the optimal data selection strategy $\pi^*$ that is capable of selecting a subset $\mathcal{S}_\pi$ maximizing the performance of the fine-tuned LLM $f_\theta$ on the downstream tasks $\mathcal{D}_{\text{tst}}$. Therefore, the data selection problem can be formally formulated as:

$$\pi^* = \arg\max_\pi \mathcal{R}\left(f_\theta; \mathcal{D}_{\text{tst}}\right), \text{ subject to } |\mathcal{S}_\pi| = k,$$
$$\text{where } \theta = \text{FineTune}(\mathcal{F}, \mathcal{S}_\pi), \tag{1}$$

where $\mathcal{S}_\pi$ is the subset of the training data selected by the strategy $\pi$, $\theta = \text{FineTune}(\mathcal{F}, \mathcal{S}_\pi)$ denotes the parameters of the model backbone $\mathcal{F}$ after fine-tuning on the selected data subset $\mathcal{S}_\pi$, $f_\theta$ is the fine-tuned model with parameters $\theta$, and $\mathcal{R}\left(f_\theta; \mathcal{D}_{\text{tst}}\right)$ is the performance metric (e.g., accuracy) of the fine-tuned model $f_\theta$ evaluated on the test set $\mathcal{D}_{\text{tst}}$.

### 3.2. Modeling Datasets as Bipartite Graphs

In our approach, we model the dataset as a bipartite graph to effectively represent the relationships between sentences and their constituent n-grams. A bipartite graph is a special type of graph whose vertices can be divided into two disjoint and independent sets such that every edge connects a vertex from one set to a vertex from the other set. Formally, a bipartite graph $\mathcal{G} = (\mathcal{U}, \mathcal{V}, \mathcal{E})$ consists of *sentence nodes* ($\mathcal{U} = \{u_i\}_{i=1}^N$), *n-gram nodes* ($\mathcal{V} = \{v_j\}_{j=1}^M$), and *edges* ($\mathcal{E} \subseteq \mathcal{U} \times \mathcal{V}$). This structure allows us to capture the occurrence of n-grams within sentences, providing a foundation for selecting sentences that maximize n-gram coverage while adhering to specific priorities. We introduce the details of the priority for re-ranking the sentences based on both quality and diversity in Section 3.3.

**GRAPHFILTER** Our objective is to select a subset of sentences, denoted as $\mathcal{S}$, from the entire dataset, constrained by a data selection budget $k$. The aim is to maximize the coverage of unique n-grams while aligning with a priority function $\phi(u)$ for each sentence $u \in \mathcal{U}$. As illustrated in Algorithm 1, our method, referred to as GRAPHFILTER, operates iteratively by updating the graph structure to reflect the n-gram coverage as sentences are selected. The process begins with an empty set of selected sentences, $\mathcal{S} = \emptyset$, and a bipartite graph $\mathcal{G}$ that includes sentence nodes, n-gram nodes, and connecting edges. In each iteration, we select the sentence $u^* \in \mathcal{U}$ that has the highest priority score $\phi(u^*)$, add $u^*$ to $\mathcal{S}$, and then remove $u^*$ from the set of remaining sentences $\mathcal{U}$. Next, we identify the n-grams covered by $u^*$, denoted as $\mathcal{V}_{u^*}$. We then remove all edges that connect $u^*$ to the n-gram nodes in $\mathcal{V}_{u^*}$. Subsequently, all edges connecting to nodes in $\mathcal{V}_{u^*}$ are eliminated from the graph. Note that the priority of each sentence $u \in \mathcal{U}$ is computed based on the most recent graph $\mathcal{G}$ during each iteration.

**Set Cover Problem** Our problem formulation is related to the classical *set cover NP-hard problem* (Garey & Johnson, 1979). In the set cover problem, given a universe of elements and a collection of sets whose union comprises the universe, the objective is to identify the smallest number of sets whose union still contains all elements in the universe. Similarly, in a special case of our problem where the priority function assigns the same score to all sentences (i.e., $\phi(u) = 1$ for all $u \in \mathcal{U}$), and the goal is to find the minimal set of sentences that cover *all* n-grams, our task becomes analogous to the set cover problem. In this scenario, the *greedy approach* used in Algorithm 1 can be shown to have an approximation factor of $H(r)$ (Vazirani, 2001), where $r$ is the maximum degree of the sentence nodes in the graph (the largest number of n-grams contained in any sentence), and $H(r) = \sum_{k=1}^r \frac{1}{k}$ is the $r$-th harmonic number. This relationship highlights the theoretical foundations of our method and provides insight into its performance guarantees in this special case.

**A Minimalist Example** Moreover, we present a minimalist example in Figure 1. Initially, the bipartite graph is displayed in Figure 1a. In Figure 1b, the sentence node $u_1$ is selected as $u^*$ in Algorithm 1, along with its associated

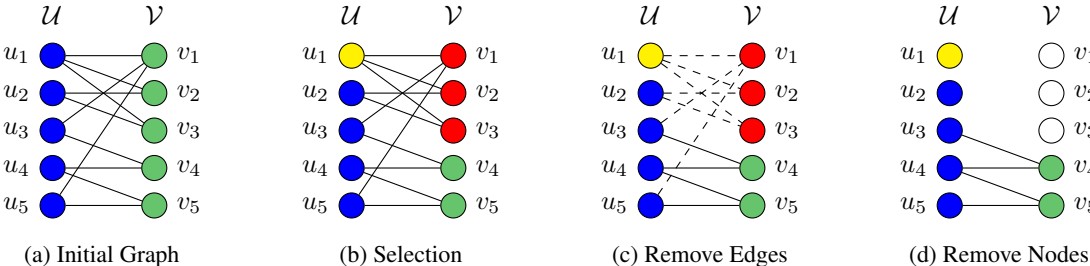

| $\mathcal{U}$ $\mathcal{V}$ | $\mathcal{U}$ $\mathcal{V}$ | $\mathcal{U}$ $\mathcal{V}$ | $\mathcal{U}$ $\mathcal{V}$ |
|---|---|---|---|
| (a) Initial Graph | (b) Selection | (c) Remove Edges | (d) Remove Nodes |

Figure 1: An example of a single iteration of GRAPHFILTER without the priority function. In this case, the degree of a sentence node serves as the priority score. Sentence nodes are in blue and n-gram nodes in green. The selected sentence node is yellow, while connected n-gram nodes are red. Removed n-gram nodes are white, with removed edges as dashed lines. Node $u_1$ is selected in the current iteration, and $u_4$ will be the next.

n-gram nodes, $\mathcal{V}_{u^1}$, which are highlighted in red. Figure 1c demonstrates the removal of edges connected to $u_1$ and $\mathcal{V}_{u^1}$, as indicated by dashed lines. Finally, Figure 1d illustrates the removal of isolated nodes, shown in white. The next selected sentence node is $u_4$. In this example, GRAPHFILTER can cover all the n-grams by selecting only $u_1$ and $u_4$.

By modeling the dataset as a bipartite graph and employing an iterative selection algorithm, GRAPHFILTER effectively selects a subset of sentences that maximizes n-gram coverage while adhering to specified priorities. *Note that each SFT training instance comprises instructions and responses. In this work, we apply* GRAPHFILTER *solely to the instructions of the SFT data.*

**Implementation** In a brute-force implementation, the computational complexity of our algorithm is $\mathcal{O}(N)$ per iteration. This complexity results from the need to perform operations such as selecting the highest-priority sentence and removing edges, which involve scanning the sets of sentences ($\mathcal{U}$), n-grams ($\mathcal{V}$), and edges ($\mathcal{E}$). These sets are not optimized for efficient access or modification. To enhance computational efficiency, we employ a max-heap (or priority queue) to select the highest-priority sentence, allowing this selection to be performed in $\mathcal{O}(\log N)$ time per iteration. This reduces the selection complexity from $\mathcal{O}(N)$ to $\mathcal{O}(\log N)$. Additionally, the max-heap data structure facilitates the localization of priority updates to affected nodes, eliminating the need to enumerate all nodes and edges.

### 3.3. Balancing Quality and Diversity with Priority Function

As illustrated in Algorithm 1, GRAPHFILTER naturally selects a subset with maximal n-gram coverage, emphasizing data diversity. However, the quality of the data is equally important for effective language model training. To balance both quality and diversity in our selection process, we define a priority function $\phi(u)$ for each sentence node $u \in \mathcal{U}$, which is used to re-rank the sentence nodes during selection.

**Quality Metric** For quality, we employ the SUPERFILTER as the quality measure (Li et al., 2024a;b). The SUPERFILTER metric evaluates the informativeness of a response by comparing the perplexity of the response conditioned on the instruction with the perplexity of the response alone. Formally, for a given sentence node $u$ associated with the instruction-response pair $(x, y)$, the quality priority metric is defined as:

$$\text{QUALITY}(u) = \text{SUPERFILTER}(x, y) = \frac{\text{PPL}(y \mid x)}{\text{PPL}(y)},$$

$$\text{where } \text{PPL}(\boldsymbol{w}) = \exp\left(-\frac{1}{T}\sum_{t=1}^{T}\log P(w_t \mid \boldsymbol{w}_{<t})\right),$$

$$(2)$$

where $\text{PPL}(\boldsymbol{w})$ is the perplexity of the sentence $\boldsymbol{w}$ with a length of $T$, $\text{PPL}(y)$ is the perplexity of the response $y$, and $\text{PPL}(y \mid x)$ is the perplexity of the response $y$ conditioned on the instruction $x$. A higher SUPERFILTER value indicates that the response is more relevant and informative given the instruction, thus reflecting higher quality. *It is important to note that the choice of quality metric can be determined based on specific user needs and the quality scores can be precomputed prior to the selection process.*

**Diversity Metric** For diversity, we use the Term Frequency-Inverse Document Frequency (TF-IDF) as a measure of the significance of each n-gram within the dataset. The TF-IDF score of an n-gram $v$ is calculated as $\text{TF-IDF}(v) = \text{TF}(v) \times \text{IDF}(v)$, where $\text{TF}(v)$ (Term Frequency) is the number of times n-gram $v$ appears in the sentence, and $\text{IDF}(v)$ (Inverse Document Frequency) is defined as $\text{IDF}(v) = \log\left(\frac{N}{d_v}\right)$, with $N$ being the total number of sentences in the corpus, and $d_v$ being the number of sentences containing n-gram $v$. Furthermore, we compute the sum of TF-IDF scores of all n-grams (of varying lengths) present in the sentence:

$$\text{DIVERSITY}(u) = \sum_{v \in \mathcal{V}_u} \text{TF-IDF}(v), \qquad (3)$$

where $\mathcal{V}_u$ is the set of n-grams connected to sentence $u$ in the graph $\mathcal{G}$. *In our work, $\mathcal{V}_u$ includes unigrams ($n = 1$), bigrams ($n = 2$), and trigrams ($n = 3$) present in sentence $u$, capturing both word-level and phrase-level features.*

**Priority Function** To effectively prioritize sentences based on both quality and diversity, we combine the QUALITY score and the DIVERSITY score for the sentence node $u$ into a single priority function:

$$\phi(u) = \text{QUALITY}(u) \times \text{DIVERSITY}(u). \quad (4)$$

This function assigns higher priority to sentences that are both high-quality and contribute significantly to n-gram diversity. By integrating both quality and diversity into the priority function, our selection algorithm can effectively choose a subset of examples that are both high-quality and broadly representative of the entire dataset.

## 4. Experiments

In this section, we initially outline our experimental setup in Section 4.1, followed by our main results in Section 4.2.

### 4.1. Experimental Setup

**Training Dataset** Xu et al. (2024) utilize state-of-the-art open-source large language models (LLMs) to create a high-quality dataset collection known as `Magpie`. In our research, we employ the `Magpie` dataset, which is generated by LLAMA-3-70B-INSTRUCT and comprises 300K training instances.[1] *For this study, we choose a subset of 10K training instances using various selection methods from the entire dataset, unless otherwise stated.*

**Baselines** We compare our approach, GRAPHFILTER, with a diverse array of baseline methods:

- **Heuristic**: (1) RANDOM randomly selects a subset from the entire dataset; (2) LONGEST sorts the training instances in descending order based on the length of the instructions;
- **Quality-based**: (3) PERPLEXITY utilizes perplexity values, where larger values typically indicate higher difficulty and quality of training instances; (4) AR-MORM is the state-of-the-art open-sourced reward model presented by Wang et al. (2024); (5) ALPAGA-SUS demonstrates that state-of-the-art LLMs can be directly prompted for estimating data quality (Chen et al., 2024b); (6) DEITA leverages CHATGPT to synthesize a quality estimation dataset and fine-tune LLMs for data quality estimation (Liu et al., 2024); (7) SUPER-FILTER indicates the Instruction-Following Difficulty

(IFD) metric computed by smaller language models. Li et al. (2024b) introduce this method, while Li et al. (2024a) demonstrate that smaller models can be used for computing IFD scores.
- **Diversity-based**: (8) KMEANS clusters the training instances with the state-of-the-art sentence embedding model and selects the training instances that are closest to their respective cluster centroids (Arthur & Vassilvitskii, 2007); (9) INSTAG is designed for analyzing the SFT dataset by tagging the topics of training instances, and can be used for selecting the subset with the most diverse topics from the entire dataset (Lu et al., 2024).

We present more details of baselines in Section A.1. We conduct experiments using three diverse model backbones, including GEMMA-2-2B (Team et al., 2024), MISTRAL-7B-v0.3 (Jiang et al., 2023), and LLAMA-3-8B (Dubey et al., 2024). The optimization details are in Section A.2.

**Evaluation** We conduct evaluations on six popular benchmarks, categorized into two groups:

- **Standardized**: We assess the LLMs using LM-EVALUATION-HARNESS (Gao et al., 2024) on four standardized benchmarks: `MMLU` (Hendrycks et al., 2021), `ARC` (Clark et al., 2018), `HellaSwag` (Zellers et al., 2019), and `GSM8K` (Cobbe et al., 2021). The model performance on these benchmarks is measured by accuracy. We use the macro-average accuracy across four benchmarks as the overall performance of this group, denoted as $\mu_{\text{BENCH}}$.
- **LLM-as-a-Judge**: We evaluate LLMs using `AlpacaEval-2.0` (Dubois et al., 2024) and `MT-Bench` (Zheng et al., 2023), with GPT-4O-2024-05-13 as the judge. For `AlpacaEval-2.0`, GPT-4-1106-PREVIEW generates reference answers, and we report both the length-controlled win rate (LC) and the original win rate (WR). For `MT-Bench`, performance is denoted as $\mu_{\text{MT}}$, the macro-average across all categories. Overall performance of this group, $\mu_{\text{LLM}}$, is the macro-average of LC and $\mu_{\text{MT}}$.

We define overall model performance, $\mu_{\text{ALL}}$, as the macro-average of results from four standardized benchmarks, LC, and $\mu_{\text{MT}}$. In calculating $\mu_{\text{ALL}}$ and $\mu_{\text{LLM}}$, $\mu_{\text{MT}}$ is scaled by $10\times$ to align with a range of 1 to 100, matching other benchmarks. Further evaluation details are in Section A.3.

### 4.2. Main Results

**GRAPHFILTER surpasses all baseline approaches.** As shown in Table 1, GRAPHFILTER consistently outperforms all baseline approaches across the three model backbones on both standardized benchmarks and LLM-as-a-Judge benchmarks. It achieves either the best or second-best results on most individual benchmarks. Specifically, in terms of $\mu_{\text{ALL}}$,

---

[1] https://huggingface.co/datasets/Magpie-Align/Magpie-Pro-300K-Filtered

Table 1: Main results given by GEMMA-2-2B, MISTRAL-7B-V0.3, and LLAMA-3-8B on the standardized benchmarks and LLM-as-a-Judge benchmarks. HS, G8K, and AE-2 correspond to HellaSwag, GSM8K, and AlpacaEval-2.0, respectively. The best results are highlighted in **bold**, and the second-best results are highlighted in underline.

| | Standardized | | | | | LLM-as-a-Judge | | | | | | $\mu_{\text{ALL}}$ |
| | MMLU | ARC | HS | G8K | $\mu_{\text{BENCH}}$ | AE-2 | | MT-Bench | | | $\mu_{\text{LLM}}$ | |
| | | | | | | LC | WR | $\mu_{\text{MT}}$ | 1st | 2nd | | |
| | Acc | Acc | Acc | Acc | | | | | | | | |
| GEMMA-2-2B | | | | | | | | | | | | |
| RANDOM | 25.25 | 47.52 | 58.27 | 9.10 | 35.03 | 10.77 | 13.73 | 4.73 | 5.44 | 4.01 | 29.01 | 33.03 |
| LONGEST | 25.50 | 47.06 | 56.43 | 8.79 | 34.45 | 10.40 | 13.10 | 4.79 | 5.54 | 4.05 | 29.17 | 32.69 |
| PERPLEXITY | 23.34 | 47.58 | 59.04 | 6.48 | 34.11 | 12.19 | 14.76 | 4.98 | 5.75 | 4.21 | 31.00 | 33.07 |
| ARMORM | 25.42 | **48.06** | 56.19 | 10.62 | 35.07 | **13.40** | **16.39** | 4.84 | 5.55 | 4.14 | 30.92 | 33.69 |
| ALPAGASUS | 26.56 | 47.18 | 58.69 | 10.57 | 35.75 | 13.12 | 15.76 | 4.89 | 5.68 | 4.11 | 31.03 | 34.18 |
| DEITA | 28.72 | 47.51 | 58.35 | 10.16 | 36.18 | 12.99 | 15.86 | 4.82 | 5.62 | 4.01 | 30.57 | 34.31 |
| SUPERFILTER | 28.82 | 47.20 | 59.18 | 9.33 | 36.13 | 12.87 | 15.55 | 4.88 | 5.49 | 4.26 | 30.81 | 34.36 |
| KMEANS | 28.39 | 46.96 | 56.59 | 10.31 | 35.56 | 12.19 | 14.76 | 4.98 | 5.74 | 4.23 | 31.00 | 34.04 |
| INSTAG | 27.60 | 47.75 | **59.98** | 9.86 | 36.29 | 12.75 | 15.47 | 4.79 | 5.45 | 4.13 | 30.31 | 34.30 |
| GRAPHFILTER | **29.06** | 47.92 | 59.38 | 10.71 | **36.77** | 13.14 | 15.99 | **5.01** | 5.77 | 4.25 | **31.64** | **35.06** |
| MISTRAL-7B-V0.3 | | | | | | | | | | | | |
| RANDOM | 25.50 | 52.17 | 67.44 | 9.17 | 38.57 | 14.76 | 17.41 | 5.03 | 5.93 | 4.13 | 32.51 | 36.55 |
| LONGEST | 25.17 | 52.11 | 67.32 | 10.30 | 38.73 | 13.67 | 16.14 | 4.96 | 6.00 | 3.91 | 31.62 | 36.36 |
| PERPLEXITY | 30.64 | 52.42 | 69.31 | 4.62 | 39.25 | 13.60 | 16.18 | 4.98 | 6.01 | 3.95 | 31.70 | 36.73 |
| ARMORM | 28.84 | 50.85 | 68.85 | 9.63 | 39.54 | **15.56** | **18.89** | 5.13 | 5.93 | 4.34 | 33.43 | 37.51 |
| ALPAGASUS | 28.67 | 51.92 | 68.61 | 9.48 | 39.67 | 14.67 | 18.14 | 5.21 | 6.13 | 4.30 | 33.40 | 37.58 |
| DEITA | 29.86 | 50.82 | 67.99 | 10.60 | 39.82 | 14.08 | 16.49 | 5.03 | 5.93 | 4.13 | 32.18 | 37.27 |
| SUPERFILTER | **33.59** | 52.45 | 68.56 | 9.93 | 41.13 | 13.59 | 16.75 | 5.23 | 6.01 | 4.44 | 32.92 | 38.40 |
| KMEANS | 28.77 | 50.58 | 67.81 | 11.55 | 39.68 | 13.98 | 16.83 | 5.11 | 5.93 | 4.29 | 32.52 | 37.29 |
| INSTAG | 28.29 | 50.99 | 67.44 | **12.59** | 39.82 | 14.55 | 17.36 | 5.11 | 5.86 | 4.36 | 32.84 | 37.50 |
| GRAPHFILTER | 33.24 | 52.48 | **69.69** | 11.92 | **41.83** | 15.16 | 18.85 | **5.38** | **6.23** | **4.54** | **34.49** | **39.38** |
| LLAMA-3-8B | | | | | | | | | | | | |
| RANDOM | 49.55 | 52.00 | 67.30 | 22.14 | 47.75 | 22.17 | 25.05 | 5.99 | 6.95 | 5.03 | 41.04 | 45.51 |
| LONGEST | 44.52 | 50.56 | 67.99 | 24.56 | 46.91 | 20.17 | 22.67 | 5.97 | 6.82 | 5.13 | 39.96 | 44.59 |
| PERPLEXITY | 51.08 | 52.31 | **68.74** | 20.96 | 48.27 | 20.38 | 22.87 | 6.02 | 7.02 | 5.01 | 40.28 | 45.61 |
| ARMORM | 47.84 | 52.24 | 68.11 | 24.64 | 48.21 | **23.45** | 26.60 | 6.19 | 7.14 | 5.24 | 42.66 | 46.36 |
| ALPAGASUS | 49.90 | 51.63 | 68.40 | 25.89 | 48.96 | 22.90 | 25.94 | 6.09 | 7.05 | 5.13 | 41.90 | 46.60 |
| DEITA | 48.49 | 52.40 | 68.46 | 25.78 | 48.78 | 22.23 | 24.42 | 6.12 | 7.12 | 5.11 | 41.70 | 46.42 |
| SUPERFILTER | 50.16 | 51.10 | 67.70 | 27.45 | 49.10 | 22.54 | 24.68 | 6.13 | **7.23** | 5.03 | 41.91 | 46.70 |
| KMEANS | 51.98 | 51.35 | 67.15 | 25.12 | 48.90 | 22.06 | 24.80 | 6.14 | 7.03 | 5.25 | 41.72 | 46.51 |
| INSTAG | 53.16 | 52.85 | 67.86 | 25.85 | 49.93 | 22.10 | 24.64 | 6.13 | 7.05 | 5.21 | 41.72 | 47.19 |
| GRAPHFILTER | **53.73** | **52.92** | 67.76 | **27.81** | **50.55** | 22.95 | **26.71** | **6.26** | 7.21 | **5.31** | **42.79** | **47.97** |

GRAPHFILTER outperforms the baselines by up to $+2.37$ for GEMMA-2-2B, $+3.02$ for MISTRAL-7B-V0.3, and $+3.38$ for LLAMA-3-8B, compared to LONGEST. These results demonstrates the superiority of GRAPHFILTER which effectively combines the quality and diversity in selection.

**Quality-based data selection approaches appear to exhibit biases towards specific benchmarks.** Quality-based approaches often use neural models to estimate the quality of each training instance. However, these models display biases that can significantly affect downstream performance. As demonstrated in Table 1, models fine-tuned on subsets chosen by ARMORM perform well on AlpacaEval-2.0 but poorly on other benchmarks. Furthermore, the PERPLEXITY-selected subset consistently re-

sults in the worst performance on GSM8K, highlighting the risks of depending solely on neural models for selecting high-quality data.

**GRAPHFILTER is highly efficient, with its variant running quickly on a CPU.** Recent baselines typically rely on neural models for quality estimation, which generally require a GPU. We compare the runtimes of various baselines on a system equipped with an A100 80G GPU and 20 CPU cores, as shown in Table 2. As elaborated in Section 3.3, GRAPHFILTER defaults to using a quality estimation model for QUALITY($u$). When utilizing SUPERFILTER, GRAPHFILTER completes its tasks in 2.48 hours, highlighting its efficiency. Notably, without using the priority function $\phi(u)$ for re-ranking, GRAPHFILTER becomes even faster, taking

Table 2: Runtime (in hours) for selecting 10K training instances. † indicate the CPU-only method.

| | Runtime (hrs) |
|---|---|
| PERPLEXITY | 0.92 |
| ARMORM | 5.93 |
| ALPAGASUS | 32.34 |
| DEITA | 22.65 |
| SUPERFILTER | 1.95 |
| KMEANS | 2.26 |
| INSTAG | 25.48 |
| GRAPHFILTER | 2.48 |
| w/o priority $\phi(u)$ | $0.53^{\dagger}$ |

Table 3: Ablation study for n-gram combination with LLAMA-3-8B. ✓ indicates that various n-grams are used.

| N-gram | | | $\mu_{\text{BENCH}}$ | $\mu_{\text{LLM}}$ | $\mu_{\text{ALL}}$ |
|---|---|---|---|---|---|
| Unigram | Bigram | Trigram | | | |
| ✓ | ✓ | ✓ | 50.55 | 42.79 | 47.97 |
| ✓ | | | 49.02 | 41.41 | 46.48 |
| | ✓ | | 49.09 | 41.70 | 46.63 |
| | | ✓ | 49.84 | 41.78 | 47.15 |

only 0.53 hours on a CPU. This is up to $61\times$ faster than other baselines, compared to 32.34 hours by ALPAGASUS.

## 5. Analysis

**Combining n-grams captures features at different levels.** We examine the effectiveness of n-gram combinations, which are designed to capture both word-level and phrase-level features. The results are presented in Table 3. Our observations suggest that the variant of GRAPHFILTER, which integrates unigrams ($n = 1$), bigrams ($n = 2$), and trigrams ($n = 3$), significantly outperforms other variations that do not incorporate n-gram combinations. Different n-grams capture features at varying levels, and merging them can effectively consolidate this information.

**Both QUALITY($u$) and DIVERSITY($u$) in priority function enhance the data selection.** We provide empirical evidence in Table 4 showcasing the effectiveness of our proposed priority function. By incorporating the QUALITY($u$) metric (using SUPERFILTER) and the DIVERSITY($u$) metric (using TF-IDF) into GRAPHFILTER, we achieve superior performance across all evaluation metrics. This demonstrates that our combined priority function significantly enhances the model's ability to select high-quality and diverse training data. Omitting either the quality metric (✗ + TF-IDF) or the diversity metric (SUPERFILTER + ✗) results in noticeable performance declines. Furthermore, replacing the SUPERFILTER metric with PERPLEXITY as the quality

Table 4: Ablation study for QUALITY($u$) and DIVERSITY($u$) in the priority with LLAMA-3-8B. ✗ indicates the component is not used.

| QUALITY($u$) | DIVERSITY($u$) | $\mu_{\text{BENCH}}$ | $\mu_{\text{LLM}}$ | $\mu_{\text{ALL}}$ |
|---|---|---|---|---|
| RANDOM | TF-IDF | 47.75 | 41.04 | 45.51 |
| SUPERFILTER | TF-IDF | 50.55 | 42.79 | 47.97 |
| PERPLEXITY | TF-IDF | 49.21 | 40.85 | 46.43 |
| ARMORM | TF-IDF | 49.01 | 41.85 | 46.61 |
| DEITA | TF-IDF | 49.11 | 41.97 | 46.73 |
| ✗ | TF-IDF | 48.94 | 41.87 | 46.58 |
| SUPERFILTER | ✗ | 49.52 | 41.28 | 46.78 |
| ✗ | ✗ | 48.27 | 40.28 | 45.61 |
| SUPERFILTER | MTLD | 50.15 | 42.42 | 47.47 |
| SKYWORKRM | MTLD | 49.51 | 42.01 | 46.93 |

Table 5: Ablation study for the choice of n-gram order with LLAMA-3-8B. #nodes indicates the total number of nodes in the bipartite graph. RT indicates the runtime in hours.

| n-gram | #nodes | RT (hrs) | $\mu_{\text{BENCH}}$ | $\mu_{\text{LLM}}$ | $\mu_{\text{ALL}}$ |
|---|---|---|---|---|---|
| 1 | 0.1M | 2.12 | 49.02 | 41.41 | 46.48 |
| 2 | 1.0M | 2.30 | 49.58 | 42.14 | 47.31 |
| 3 | 2.6M | 2.48 | 50.55 | 42.79 | 47.97 |
| 4 | 4.8M | 3.38 | 50.11 | 42.63 | 47.43 |
| 5 | 7.4M | 4.58 | 50.44 | 42.81 | 47.95 |

measure leads to reduced performance, highlighting the importance of using optimal metrics. These findings support our decision to integrate quality and diversity in the priority.

**GRAPHFILTER is compatible with various quality metrics.** In this work, we leverage SUPERFILTER as the default quality metric due to its effectiveness and efficiency. However, GRAPHFILTER is designed to be flexible and is not limited to using SUPERFILTER alone. This flexibility allows the method to adapt to different evaluation needs by incorporating alternative quality metrics. As demonstrated in Table 4, the QUALITY($u$) component of the priority function $\phi(u)$ can be replaced with various quality metrics, such as PERPLEXITY, ARMORM, and DEITA. This adaptability highlights the robustness and versatility of GRAPHFILTER.

**The choice of trigrams ($n = 3$) balances the model performance and efficiency.** We conduct experiments with n-gram sizes from 1 to 5 using LLAMA-3-8B. As shown in Table 5, our results indicate a significant performance improvement when moving from unigrams ($n = 1$) to trigrams ($n = 3$). However, beyond $n = 3$, we observe diminishing or even negative returns. Furthermore, the number of n-gram nodes increases substantially with n (from 0.1M for unigrams to 7.4M for 5-grams), as well as the runtime (from 2.12 hours to 4.58 hours). These findings demonstrate that the trigrams ($n = 3$) is the optimal choice for balancing performance and efficiency in our experiments.

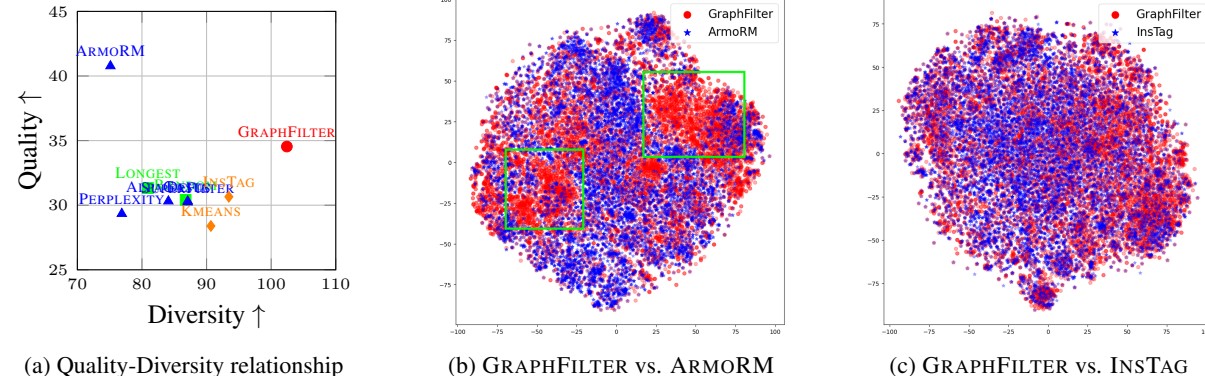

(a) Quality-Diversity relationship     (b) GRAPHFILTER vs. ARMORM     (c) GRAPHFILTER vs. INSTAG

Figure 2: Figure 2a displays the quality-diversity relationships of subsets selected by different methods, with ↑ indicating a preference for higher values. Figure 2b shows the semantic diversity in a t-SNE plot of subsets from GRAPHFILTER and ARMORM, where green rectangles indicate data points chosen by GRAPHFILTER but not by ARMORM. Figure 2c depicts the semantic diversity in a t-SNE plot comparing subsets from GRAPHFILTER and INSTAG.

Table 6: Applying GRAPHFILTER to instructions and responses with LLAMA-3-8B. The ✓ indicates that GRAPHFILTER is applied. Lexical diversity is measured by MTLD (McCarthy & Jarvis, 2010), and quality is assessed using ARMORM, scaled by $100\times$.

| Content Type | | Benchmarks | | | Lexical Diversity | | Quality |
|---|---|---|---|---|---|---|---|
| Inst. | Resp. | $\mu_{\text{BENCH}}$ | $\mu_{\text{LLM}}$ | $\mu_{\text{ALL}}$ | Inst. | Resp. | |
| ✓ | | 50.55 | 42.79 | 47.97 | 102.43 | 71.74 | 81.54 |
| | ✓ | 47.16 | 39.71 | 44.68 | 90.22 | 73.57 | 81.52 |
| ✓ | ✓ | 48.03 | 41.20 | 45.76 | 90.13 | 72.60 | 81.52 |

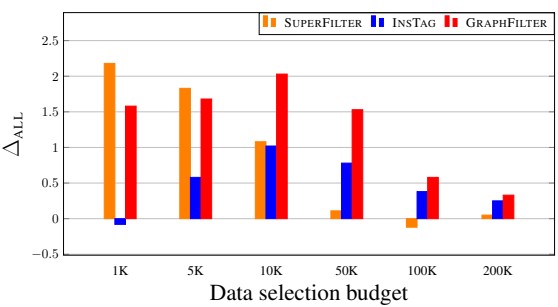

Figure 3: Performance gap ($\Delta_{\text{ALL}}$) with respect to $\mu_{\text{ALL}}$, comparing SUPERFILTER, INSTAG, and GRAPHFILTER against RANDOM, across various data selection budgets.

**GRAPHFILTER effectively balances quality and diversity in its selected datasets.** In this section, we analyze the subsets selected by GRAPHFILTER and other methods, with results shown in Figure 2. To confirm that GRAPHFILTER maintains quality and diversity, we measure lexical diversity using the MTLD metric (McCarthy & Jarvis, 2010) and assess data quality with the advanced reward model, SKYWORKRM (Liu & Zeng, 2024). As depicted in Figure 2a, GRAPHFILTER achieves the highest lexical diversity and ranks second in data quality. We also visualize GRAPHFILTER instructions compared with ARMORM and INSTAG using the BGE-LARGE-EN-V1.5 model. It is evident that GRAPHFILTER selects instructions not chosen by ARMORM, shown by green rectangles in Figure 2b. Furthermore, Figure 2c illustrates that GRAPHFILTER and INSTAG exhibit similar semantic diversity. These results suggest that GRAPHFILTER not only selects high-quality data but also maximizes dataset diversity.

**Prioritizing instruction diversity most effectively improves model performance.** Each SFT training instance comprises an instruction and its response. This study evalu-

ates the impact of applying GRAPHFILTER to instructions, responses, or both on model performance. As shown in Table 6, applying GRAPHFILTER only to instructions produces the best benchmark results, greatly improving lexical diversity in instructions with minimal effect on response diversity compared to other methods. Notably, all three variations maintain similar quality with different performances, underscoring the importance of instruction diversity.

**The priority of quality and diversity varies with data selection budgets, and GRAPHFILTER excels at balancing these two factors effectively.** After showcasing GRAPHFILTER's superiority, an open question remains: *When should diversity be prioritized over quality, and vice versa?* We hypothesize that the data selection budget plays a crucial role in determining the priority between quality and diversity and present the results in Figure 3. Our results indicate that the effectiveness of quality-based and diversity-based strategies is budget-dependent. Specifically, the quality-based SUPERFILTER excels with smaller budgets (1K and

5K instances), but its advantage diminishes as the budget increases. This suggests that quality-based methods with neural models may exhibit biases toward certain linguistic patterns, which limits model generalization when the budget is sufficiently large. Conversely, the diversity-based INSTAG performs poorly with small budgets but surpasses SUPER-FILTER with larger ones. This observation demonstrates that diversity-based methods are more prone to introducing low-quality data with smaller budgets. Notably, GRAPHFILTER consistently achieves significant performance gains compared to RANDOM across all budget levels. These findings show that the data selection budget influences the effectiveness of different approaches, and GRAPHFILTER successfully integrates both quality and diversity.

## 6. Conclusion

In this work, we formulate data selection as a set cover problem and introduce GRAPHFILTER, a novel method for data selection that models the dataset as a bipartite graph linking sentences to their constituent n-grams. To balance quality and diversity, we use a priority function that combines a quality metric with a diversity metric, allowing us to select subsets that enhance n-gram diversity and maintain high response quality. Our extensive experiments demonstrate GRAPHFILTER's effectiveness across three model backbones and six benchmark datasets. Compared to nine baseline methods, GRAPHFILTER consistently delivers superior model performance and computational efficiency. Our analyses validate our design choices, assess the subsets chosen by GRAPHFILTER and other methods, highlight the importance of instruction diversity, and examine the role of quality and diversity relative to subset sizes. We believe GRAPHFILTER lays the groundwork for more effective data selection strategies, encouraging further research in data selection for LLMs.

## Impact Statement

Our proposed method, GRAPHFILTER, aims to improve the efficiency and effectiveness of LLM training by enabling the selection of high-quality, diverse data subsets. This could lead to more resource-efficient model training processes and potentially better-performing models. While the primary impact of our work is methodological, we acknowledge that improvements in LLM training techniques could indirectly contribute to the broader adoption and application of these models across society. We believe the computational efficiency benefits of our approach align with efforts to reduce the environmental impact of training large models. As with any advancement in AI capabilities, we encourage thoughtful consideration of how improved language models might be deployed and used in practice.

## Acknowledgements

We would like to thank the anonymous reviewers and program chairs for their constructive feedback and valuable suggestions, which have greatly improved the quality of this paper. This work is partly supported by the ARC Future Fellowship FT190100039.

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

# A. Experimental Setup

## A.1. Baselines

In this work, we compare GRAPHFILTER against following baselines:

- **RANDOM** selects a random subset of size $k$ from the entire dataset, where $k$ is the designated data selection budget.
- **LONGEST** chooses the top-$k$ instances from the entire dataset, ranking them in descending order based on the number of words in each instruction.
- **PERPLEXITY** selects the top-$k$ instances from the entire dataset, sorted in descending order according to the perplexity values of the instructions. For the perplexity computation in this work, we utilize GPT2 (Radford et al., 2019).[2]
- **ARMORM** represents one of the state-of-the-art reward models (Wang et al., 2024). It evaluates multiple rewards from diverse perspectives and integrates these rewards using a gating network.
- **ALPAGASUS** employs GPT-3.5-TURBO to assess data quality (Chen et al., 2024b). Given the improved model performance and limited budget, we substitute GEMMA-2-27B-IT in this work, using the prompt illustrated in Figure 4. GEMMA-2-27B-IT is the state-of-the-art open large language model (LLM) and significantly surpasses GPT-3.5-TURBO according to the Chatbot Arena Leaderboard.[3]
- **DEITA** utilizes CHATGPT to create a quality estimation dataset and fine-tune large language models (LLMs) for evaluating data quality (Liu et al., 2024). We employ the official codes and models provided by Liu et al. (2024) for data selection.[4]
- **SUPERFILTER** refers to the Instruction-Following Difficulty (IFD) metric, which is calculated using smaller language models. Introduced by Li et al. (2024b), this method is shown by Li et al. (2024a) to provide IFD scores from smaller models that are as reliable as those from larger models. In this study, GPT2 is used for computing these scores (Radford et al., 2019).
- **KMEANS** involves clustering training instances using a state-of-the-art sentence embedding model and selecting instances that are nearest to their respective cluster centroids (Arthur & Vassilvitskii, 2007). In this work, we begin by sampling 50K instances from the entire dataset and encoding their instructions into sentence embeddings using the BGE-LARGE-EN-V1.5 model.[5]

---

[2]https://huggingface.co/openai-community/gpt2
[3]https://huggingface.co/spaces/lmsys/chatbot-arena-leaderboard
[4]https://github.com/hkust-nlp/deita
[5]https://huggingface.co/BAAI/

```
### System:
We would like to request your feedback on the performance of
↪  AI assistant in response to the instruction and the given
↪  input displayed following.

###Instruction:
{instruction}

### Input:
{input}

### Response:
{output}

### USER:
Please rate according to the accuracy of the response to the
↪  instruction and the input. Each assistant receives a score
↪  on a scale of 0 to 5, where a higher score indicates
↪  higher level of the accuracy. Please first output a single
↪  line containing value indicating the scores. In the
↪  subsequent line, please provide a comprehensive
↪  explanation of your evaluation, avoiding any potential
↪  bias.
```

Figure 4: The prompt used for ALPAGASUS annotation.

These embeddings are used for training the KMEANS model with 10K clusters. Once the KMEANS model is established, we cluster the sentence embeddings of instructions for the entire dataset and select the instances closest to each cluster centroid.

- **INSTAG** is designed to analyze the SFT dataset by tagging the topics of training instances. It can be used to select a subset with the most diverse topics from the entire dataset (Lu et al., 2024). We utilize the official codes and models released by Lu et al. (2024) for data selection.[6]

## A.2. Optimization

**Hyperparameters** In this study, all experiments utilize the same set of hyperparameters. Specifically, we employ a batch size of 64, a learning rate of $2 \times 10^{-5}$, a warmup ratio of 0.05, and a linear learning rate schedule. All the experiments run for 3 epochs.

**Computation Infrastructure** For this study, all methods are trained using two A100 80GB GPUs, which are interconnected via PCIe.

## A.3. Evaluation

In this work, we evaluate the approaches on six widely used benchmarks:

- **MMLU** (Hendrycks et al., 2021) is a benchmark designed to assess knowledge acquired during pretraining, by evaluating models exclusively in zero-shot and few-shot settings. It covers 57 subjects across STEM, the humanities, social sciences, and more, totaling ap-

---

bge-large-en-v1.5
[6]https://github.com/OFA-Sys/InsTag

proximately 14,000 test examples.

- **ARC** (Clark et al., 2018) is a multiple-choice question-answering dataset containing questions from science exams for grades 3 to 9, amounting to approximately 4,000 test examples.
- **HellaSwag** (Zellers et al., 2019) is a challenging dataset for evaluating commonsense natural language inference, which is particularly difficult for state-of-the-art models, though its questions are trivial for humans. It contains approximately 10,000 test examples.
- **GSM8K** (Cobbe et al., 2021) comprises a collection of diverse grade school math word problems created by human problem writers, containing approximately 1,000 test examples.
- **AlpacaEval-2.0** (Dubois et al., 2024) is an automated tool for evaluating instruction-following language models. Its test set consists of 805 instructions generated by large language models (LLMs). Models are evaluated based on the winning rate against a reference answer, judged by a state-of-the-art LLM, such as GPT-4. AlpacaEval-2.0 is an upgraded version of the original AlpacaEval, featuring reduced length bias for a fairer evaluation of responses of varying lengths.
- **MT-Bench** (Zheng et al., 2023) is a multi-turn test set containing 80 questions that cover 8 aspects: writing, roleplay, reasoning, math, coding, extraction, STEM, and humanities. A state-of-the-art LLM, such as GPT-4, is used to score model outputs on a scale from 1 to 10.

