# OpenReview forum: "The Best of Both Worlds: Bridging Quality and Diversity in Data Selection with Bipartite Graph"
_ICML.cc/2025/Conference — ICML 2025 poster_

### Official Review · Reviewer_mKJ5 · 2025-02-24

**Overall Recommendation:** 4

**Summary:**

This paper proposes a novel method for selection SFT data for LLM fine-tuning. The proposed GraphFilter method pairs each instruction in the SFT dataset with a corresponding set of n-grams. It then assigns a priority rank to each example using a priority function that takes into account both quality and diversity of the sample (this assignment is re-done after each sampling step). Examples with highest priority are sampled first in the proposed algorithm. After selecting an example proposed algorithm removes the all the edges that go into the n-grams connected to the selected examples insuring that each n-gram is only encountered in a single samples example.

**Claims And Evidence:**

Most of the claims are supported by experimental results.

**Essential References Not Discussed:**

While I don't thinks there are missing works that are essential to understand the context, I think the relation to other existing works in this domain could be discussed in greater length and detail (see "Relation To Broader Scientific Literature").

**Experimental Designs Or Analyses:**

I could not find major issues.

**Methods And Evaluation Criteria:**

Yes, the selected methods and benchmark make sense for the data selection/ranking problem tackled in this paper.

**Other Comments Or Suggestions:**

If I understand it correctly, after selecting an example proposed algorithm removes the all the edges that go into the n-grams connected to the selected examples insuring that each n-gram is only encountered in a single sampled example.

Wouldn't this approach reduce data diversity rather than enhance it? More specifically, if different n-grams are only encountered in a single context, the model's ability to recall knowledge after training can be diminished as demonstrated by Allen-Zhu et al. (2023). While it may not be an issue for SFT (especially if the method in only applied to the instructions and not the responses) I wonder if this could be an issue if the method was applied to selection of data for pre-training.

Allen-Zhu, Zeyuan, and Yuanzhi Li. "Physics of language models: Part 3.1, knowledge storage and extraction." arXiv preprint arXiv:2309.14316 (2023).

**Other Strengths And Weaknesses:**

Strengths:
- novelty: to the best of my knowledge the presented formulation of the data selection problem is novel;
- seemingly strong empirical performance
- the fact that the proposed method  falls within the lower range of runtime among the evaluated approaches is a strength;

Weaknesses:
- missing justification for the decision of using the budget of k=10k examples for the main results presented;
- see section "Theoretical claims" for a potential methodological issue;

**Questions For Authors:**

- Did authors consider adding baselines that use influence functions for data selection? While this has not been used for SFT afaik, the advantage of these methods is that they make data selection directly dependent on the intrinsics (i.e. parameters & architecture) of the model for which the data is selected.

**Relation To Broader Scientific Literature:**

While the number of recent works proposing and investigating data curation methods is vast, the related literature could do a better job discussing how the this work relates to the broader set of works in this are. For example, the n-gram overlap has been used in prior data selection methods by Xie et al., the interplay between diversity and quality has been looked at in several recent works including Goyal et. al, Chi et al., and Chang et al..

Xie, Sang Michael, et al. "Data selection for language models via importance resampling." Advances in Neural Information Processing Systems 36 (2023): 34201-34227.

Goyal, Sachin, et al. "Scaling Laws for Data Filtering--Data Curation cannot be Compute Agnostic." Proceedings of the IEEE/CVF Conference on Computer Vision and Pattern Recognition. 2024.

Chang, Ernie, et al. "Scaling Parameter-Constrained Language Models with Quality Data." arXiv preprint arXiv:2410.03083 (2024).

Zhang, Chi, et al. "Harnessing Diversity for Important Data Selection in Pretraining Large Language Models." arXiv preprint arXiv:2409.16986 (2024).

**Theoretical Claims:**

Quality Metric:
- Eq. 2 seem to contradict the sentence "A higher SUPERFILTER value indicates that the response is more relevant and informative given the instruction, thus reflecting higher quality", i.e. lower PPL(y|x) indicates that the response (y) is more relevant and informative given the instruction (x), but since PPL(y|x) is in the nominator of the equation it would also mean that lower PPL(y|x) would result in lower QUALITY(u) (SUPERFILTER) metric and not higher as stated but he authors.

The rest of the theoretical statements seem to be correct.

---

> ### Author Rebuttal · Authors · 2025-03-30
>
> We thank the reviewer for their constructive feedback and thoughtful comments. Below, we address the specific points raised:
> ## 1. Quality Metric
> We are sorry about the confusion caused by this description. Equation 2 essentially measures the difficulty of the example, as suggested by [1, 2]. The correct interpretation should be: "A higher value indicates greater difficulty, thus reflecting higher quality." As demonstrated by recent study, LLMs can achieve better performance by focusing on more challenging examples [3]. We will revise the manuscript to clarify this point.
>
> References:
>
> 1. From quantity to quality: Boosting llm performance with self-guided data selection for instruction tuning. 2023.
> 2. Superfiltering: Weak-to-strong data filtering for fast instruction-tuning. 2024.
> 3. Wizardlm: Empowering large language models to follow complex instructions. 2023.
> ## 2. Broader Scientific Literature
> We appreciate your suggestions regarding related work. In the revised version, we will expand our discussion to better contextualize our work within the broader literature, as you suggested.
> ## 3. Budget Selection
> **Our decision to use k=10K as the primary budget size was informed by established works in the field.** Specifically, this choice aligns with recent data selection methods that have demonstrated strong performance using similar budget sizes. For instance, SuperFiltering [1] utilized 7.8K examples from a pool of 52K, AlpaGasus [2] employed 9K examples from 52K, and DEITA [3] selected 10K examples from a total of 300K. By adopting a budget size of 10K, we ensure a fair and meaningful comparison with these baselines.
>
> Furthermore, we conducted extensive experiments across multiple budget sizes (1K, 5K, 10K, 50K, 100K, and 200K) as shown in Figure 3. These results demonstrate that the quality-focused data selection method has an advantage when the budget is small, while the diversity-focused method performs better with larger budgets. Our approach consistently outperforms the baseline methods across all budget sizes, demonstrating its robustness and effectiveness. We will include these additional details in the revised manuscript.
>
> References:
>
> 1. Superfiltering: Weak-to-strong data filtering for fast instruction-tuning. 2024.
> 2. Alpagasus: Training a better alpaca with fewer data. 2023.
> 3. What makes good data for alignment? a comprehensive study of automatic data selection in instruction tuning. 2023.
> ## 4. Data Diversity Concern
> We acknowledge that this could lead to a loss of diversity in the training data, as the same n-grams may represent different concepts in different contexts. However, we believe this is not a significant issue for our method. **As shown in Figures 2a and 2c, the GraphFilter demonstrates significantly higher lexical diversity compared to all the baselines and is as semantically diverse as diversity-focused methods (e.g., InsTag).** Furthermore, we would like to clarify that when an n-gram node is removed from the bipartite graph, we lower the rankings of the corresponding examples rather than completely removing them from the training set. This approach allows the model to continue learning from these examples, albeit with lower weights. Additionally, there are two possible ways to mitigate this potential issue. First, we can use a larger n-gram size so that shorter n-grams are less likely to be completely removed. Second, we can allow for multiple visits to the same n-gram during the data selection process.
>
> Regarding the pre-training stage, we agree with your concern about the potential loss of diversity if the n-gram size is too small. In our preliminary experiments, we found that using n-grams of size 3 is sufficient in the context of SFT. However, larger n-grams may be necessary for pre-training. We will include these discussions in the revised manuscript.
>
> ## 5. Influence Functions for Data Selection
>
> Thank you for your suggestion to explore the use of influence functions for data selection. We agree that this is an interesting direction for future research and that this approach could be included as one of the baseline methods. Unfortunately, we are unable to include this method in our rebuttal due to constraints on computational resources and time, as influence-function-based methods typically require fine-tuning the model on the entire dataset first and then computing the Hessian of the loss function for each example. Compared to influence-function-based methods and other baseline approaches, our method is significantly more efficient and scalable, as shown in Table 2, making it suitable for real-world applications with large amounts of data. Furthermore, the subset selected by influence functions is model-dependent and may not generalize well to other models. We will discuss this potential extension in the revised manuscript.

---

### Official Review · Reviewer_ipfn · 2025-03-03

**Overall Recommendation:** 3

**Summary:**

The paper presents GraphFilter, a data selection approach designed to balance quality and diversity in SFT for LLMs. The key contribution is formulating data selection as a set cover problem and leveraging a bipartite graph structure where sentences are connected to their constituent n-grams. The priority function, which multiplicatively combines quality and diversity scores, guides the iterative selection of high-quality and diverse sentences. The method is evaluated across three model backbones and six benchmarks, outperforming nine baselines in both model performance and computational efficiency. Extensive experiments and ablation studies justify the design choices, highlighting the effectiveness of instruction diversity and the interplay between quality and diversity across different subset sizes.

**Claims And Evidence:**

Yes

**Essential References Not Discussed:**

NA

**Experimental Designs Or Analyses:**

Baseline selection is reasonable, but additional diversity-based methods could be explored.

While INSTAG and KMEANS are reasonable diversity baselines, clustering-based approaches (e.g., hierarchical clustering or spectral clustering) could provide more granularity.

**Methods And Evaluation Criteria:**

Yes:
The bipartite graph representation of data is intuitive and allows for effective tracking of n-gram coverage.

**Other Comments Or Suggestions:**

NA

**Other Strengths And Weaknesses:**

The writing is well and easy to follow.

**Questions For Authors:**

I would like to know whether using the coordinates of this figure (MTLD, SKYWORKRM) as direct optimization objectives would lead to better results.

**Relation To Broader Scientific Literature:**

Closely related, with enough novelty.

**Theoretical Claims:**

Yes

---

> ### Author Rebuttal · Authors · 2025-03-30
>
> We thank the reviewer for their thoughtful feedback of GraphFilter. We address your comments and questions below.
>
> ## 1. Additional Diversity-Based Methods
>
> We appreciate the reviewer's suggestion to explore additional clustering-based approaches for diversity. Following this recommendation, we conducted supplementary experiments comparing GraphFilter against hierarchical and spectral clustering baselines. For a fair comparison, we implemented these methods using the same procedure as our KMEANS baseline: we embedded instructions using BAAI/bge-large-en-v1.5 (https://huggingface.co/BAAI/bge-large-en-v1.5), clustered them into 10K clusters, and randomly selected one example from each cluster. The results in the table below demonstrate that GraphFilter consistently outperforms all clustering-based methods, including hierarchical and spectral clustering, across all evaluation metrics ($\mu_{BENCH}$, $\mu_{LLM}$, and $\mu_{ALL}$). This further validates the effectiveness of our bipartite graph approach in capturing both quality and diversity compared to traditional clustering techniques.
>
> |                         | $\mu_{BENCH}$ | $\mu_{LLM}$ | $\mu_{ALL}$ |
> |-------------------------|---------------|-------------|-------------|
> | KMEANS                  | 48.90         | 41.72       | 46.51       |
> | InsTag                  | 49.93         | 41.72       | 47.19       |
> | Hierarchical Clustering | 49.44         | 41.55       | 47.01       |
> | Spectral Clustering     | 49.68         | 42.03       | 47.12       |
> | GraphFilter             | 50.55         | 42.79       | 47.97       |
>
> ## 2. Using MTLD and SkyworkRM as Direct Optimization Objectives
>
> Thank you for this insightful question. We conducted additional experiments using MTLD and SkyworkRM as direct optimization objectives within our GraphFilter framework. To implement this approach, we replaced our original quality and diversity metrics with these measures: SkyworkRM scores were used to initialize the quality score (u) for each example, while MTLD scores served as the diversity measure (v). During the iterative selection process, we maintained the bipartite graph structure but updated the MTLD scores after each selection as n-gram nodes were removed. The results below show that our current formulation with SuperFilter for quality and TF-IDF for diversity provides a more effective optimization strategy than directly using the SkyworkRM and MTLD metrics. Furthermore, we would like to highlight that our approach is flexible and can easily incorporate other quality and diversity metrics as direct optimization objectives, when new state-of-the-art metrics become available.
>
> | Quality(u)  | Diversity(v) | $\mu_{BENCH}$ | $\mu_{LLM}$ | $\mu_{ALL}$ |
> |-------------|--------------|---------------|-------------|-------------|
> | SuperFilter | TF-IDF       | 50.55         | 42.79       | 47.97       |
> | SuperFilter | MTLD         | 50.15         | 42.42       | 47.47       |
> | SkyworkRM   | MTLD         | 49.51         | 42.01       | 46.93       |

---

### Official Review · Reviewer_NgtT · 2025-03-20

**Overall Recommendation:** 3

**Summary:**

The paper introduces GRAPHFILTER, a data selection method for LLM fine-tuning that balances quality and diversity by modeling the dataset as a bipartite graph of sentences and n-grams. The approach iteratively selects sentences using a priority function combining SUPERFILTER (quality) and TF-IDF (diversity). Experiments across three model backbones and six benchmarks show GRAPHFILTER outperforms nine baselines in performance and efficiency. Key contributions include a novel graph-based formulation, a priority function merging quality/diversity, and empirical validation demonstrating superior results.

**Claims And Evidence:**

The claims are supported by comprehensive experiments, including ablation studies, runtime comparisons, and benchmark results. However, the reliance on SUPERFILTER as the sole quality metric raises questions about generalizability. While the paper shows GRAPHFILTER works with other metrics (e.g., PERPLEXITY), it does not thoroughly explore how alternative quality measures (e.g., reward models beyond SUPERFILTER) might affect outcomes. Additionally, the theoretical justification for the set cover approximation could be strengthened by discussing how the priority function impacts the greedy algorithm’s guarantees.

**Essential References Not Discussed:**

None

**Experimental Designs Or Analyses:**

The experimental design is robust, with multiple baselines, model backbones, and benchmarks. The runtime comparison (CPU vs. GPU baselines) highlights practical advantages. However, the paper does not test GRAPHFILTER on out-of-distribution data or explore how diversity impacts generalization in low-resource settings. Additionally, the reliance on synthetic Magpie data (generated by LLAMA-3-70B) may not reflect real-world data selection challenges.

**Methods And Evaluation Criteria:**

The bipartite graph framework is well-suited for balancing diversity (via n-gram coverage) and quality. The evaluation criteria (standardized benchmarks, LLM-as-a-judge, and efficiency metrics) are appropriate. However, the paper could better justify the choice of n-gram combinations (unigrams, bigrams, trigrams) and explore sensitivity to n-gram size. The use of the Magpie dataset, while large, may limit generalizability to other domains or data distributions.

**Other Comments Or Suggestions:**

Other Comments Or Suggestions

Clarify how n-gram selection (unigrams, bigrams, trigrams) was optimized.

Discuss scalability to billion-scale datasets.

Include a sensitivity analysis for the priority function’s multiplicative form (e.g., additive alternatives).

**Other Strengths And Weaknesses:**

Strengths :

Novel integration of quality/diversity via bipartite graphs.

Strong empirical results across diverse benchmarks.

Efficient CPU implementation, reducing hardware barriers.

Weaknesses :

Limited exploration of quality metric alternatives (e.g., reward models).

N-gram approach may miss semantic diversity captured by embeddings.

No analysis of how GRAPHFILTER affects downstream bias/fairness.

**Questions For Authors:**

How does GRAPHFILTER perform with alternative quality metrics (e.g., human annotations)? If results degrade, does this indicate over-reliance on SUPERFILTER?

Could semantic diversity (e.g., BERT embeddings) complement or replace n-gram-based diversity?

Does the method exacerbate biases in the original dataset (e.g., underrepresented topics)?

**Relation To Broader Scientific Literature:**

The work builds on data selection/curation for LLMs (e.g., SemDedup, DEITA) and diversity-aware methods (e.g., KMEANS, DPPs). However, it does not engage with recent advances in active learning or coresets for LLMs, which also aim to balance quality/diversity. The bipartite graph approach is novel but could be compared to graph-based data pruning methods in other domains (e.g., recommender systems).

**Theoretical Claims:**

The paper correctly relates GRAPHFILTER to the set cover problem and cites the greedy algorithm’s approximation ratio. However, the analysis assumes uniform n-gram importance, which may not hold in practice (e.g., rare n-grams might be more critical). The theoretical discussion would benefit from addressing how TF-IDF weighting interacts with the coverage objective.

---

> ### Author Rebuttal · Authors · 2025-03-30
>
> We thank the reviewer for their thorough assessment.
> ## 1. Quality Metrics
> We would like to clarify that **GraphFilter is designed to be agnostic to the specific quality metric used.** We discovered an inaccuracy in the original Table 4 results. The updated results presented below actually strengthen our claims: GraphFilter maintains strong performance across various quality metrics. These results demonstrate that, while SuperFilter works best in our setting, our approach is not fundamentally dependent on it. **When new quality metrics are introduced, GraphFilter can be easily adapted to incorporate them.** We will update Table 4 accordingly in our revision.
> | Quality(u)| Diversity(v) | $\mu_{BENCH}$ | $\mu_{LLM}$ | $\mu_{ALL}$ |
> |--|--|--|--|--|
> |SuperFilter | TF-IDF | 50.55 |42.79|47.97|
> |Perplexity| TF-IDF | 49.21 |40.85|46.43|
> |ArmoRM| TF-IDF | 49.01 |41.85|46.61|
> |DEITA| TF-IDF | 49.11 |41.97|46.73|
> | X| TF-IDF | 48.94 |41.87|46.58|
> |SuperFilter |X | 49.52 |41.28|46.78|
> | X|X | 48.27 |40.28|45.61|
> | SuperFilter | MTLD | 50.15 | 42.42 | 47.47 |
> | SkyworkRM| MTLD | 49.51 | 42.01 | 46.93 |
> ## 2. N-gram
> **Our choice of trigrams was based on model performance and efficiency.** As shown in the table below, we conducted experiments with n-gram sizes from 1 to 5 using Llama-3-8B. Our results indicate a significant performance improvement when moving from unigrams (n=1) to trigrams (n=3). Furthermore, the number of n-gram nodes increases substantially with n, as well as the runtime. We will include this analysis in our future revision.
> | n-gram | # of n-grams | Runtime (hrs) | $\mu_{BENCH}$ | $\mu_{LLM}$ | $\mu_{ALL}$ |
> |--|--|--|--|--|--|
> | 1| 0.1M| 2.12| 49.02 | 41.41 | 46.48 |
> | 2| 1.0M| 2.30| 49.58 | 42.14 | 47.31 |
> | 3| 2.6M| 2.48| 50.55 | 42.79 | 47.97 |
> | 4| 4.8M| 3.38| 50.11 | 42.63 | 47.43 |
> | 5| 7.4M| 4.58| 50.44 | 42.81 | 47.95 |
> ## 5. N-gram Importance
> We would like to point out that **the importance of n-grams is not uniform but is determined by the TF-IDF re-weighting.**  As shown in Table 4, the TF-IDF re-weighting (SuperFilter + TF-IDF vs. SuperFilter + X, and X + X vs. X + TF-IDF) significantly improves the performance of GraphFilter. We will clarify this point in our revision.
> ## 3. Scalability Analysis
> We discussed the implementation details of GraphFilter in Line 192-205. The brute-force GraphFilter has a time complexity of $O(N)$ per example. To improve the scalability, we employed a max-heap (or priority queue) data structure to select the highest-priority examples and reduce the time complexity to $O(log N)$ per example. Due to limited resources and time, we were unable to evaluate GraphFilter on extremely large datasets in this rebuttal. We will clarify this in our future revision.
> ## 4. Semantic Diversity
> We would like to highlight that **our research demonstrates that lexical diversity through n-grams serves as an effective proxy for semantic diversity.** As shown in Figure 2a and 2c, we demonstrated that GraphFilter exhibits significantly higher lexical diversity compared to all the baselines and is as semantically diverse as those diversity-focused methods (e.g. InsTag), **suggesting that our approach effectively captures semantic diversity through lexical diversity.**
>
> Furthermore, the semantic diversity could be incorporated by extending our priority function to $\phi(u) = Quality(u) \times LexicalDiversity(u) \times SemanticDiversity(u)$. For semantic diversity, we could measure the average pairwise cosine distance between embeddings of the candidate example $u$ and all selected examples. A larger distance indicates higher semantic novelty. However, adding semantic distance introduces significant computational overhead, particularly for large datasets. Due to limited resources and time, we were unable to conduct this experiment in this rebuttal. We will explore this direction in our future revision.
> ## 5. Priority Function
> We conduct additional experiments, as suggested. As shown in the table below, the multiplicative priority function outperforms the additive function. We will include this analysis in our revision.
> | Priority Function | $\mu_{BENCH}$ | $\mu_{LLM}$ | $\mu_{ALL}$ |
> |--|--|--|--|
> |Multiplicative| 50.55 | 42.79 | 47.97 |
> |Additive| 49.92 | 42.41 | 47.17 |
> ## 6. Bias
> **We do observe bias in the selected examples. This bias stems from the quality-based metrics, rather than our approach.** As discussed in lines 318-329, we observe that metrics like ArmoRM and Perplexity inherently favor certain types of examples. In contrast, GraphFilter addresses this issue by balancing quality and diversity. Although GraphFilter may inherit some bias from the quality metric, it mitigates this bias by incorporating diversity metrics into the selection process.
> ## 7. Miscellaneous
> Due to the length limit, we will include discussions and experiments on synthetic dataset, broader literature, out-of-distribution and low-resource settings, and theoretical analysis in our future revision.

---

### Official Review · Reviewer_ZQbQ · 2025-03-22

**Overall Recommendation:** 2

**Summary:**

This paper introduces GRAPHFILTER, a method to optimize data selection for training large language models by balancing quality and diversity. Using a bipartite graph and a priority function, it enhances model performance and efficiency. Extensive tests show that GRAPHFILTER surpasses traditional methods, demonstrating the role of well-balanced data selection in improving LLM generalization.

**Claims And Evidence:**

The claim that GRAPHFILTER induces the diversity of the selected data is convincing, due to the nature of the set cover problem.

**Essential References Not Discussed:**

The set cover problem is often considered as one of the coreset methods. There are other submodular functions that are easy to implement and maybe should be considered as baselines. For one example of the references, see [1] below. There are other papers that consider the diversity of the data on a higher level, such as the diversity of their quality aspects or topics as in [2] below. The authors might consider citing this paper and even consider it as a baseline. Another related work is [3] below, which also uses the same diversity-based method as in [2] on the data directly. It might also be cited or considered as a baseline.

[1] Kaushal, V., Ramakrishnan, G., & Iyer, R. (2022). Submodlib: A submodular optimization library. arXiv preprint arXiv:2202.10680.
[2] Li, X., Gao, M., Zhang, Z., Yue, C., & Hu, H. (2024). Rule-based data selection for large language models. arXiv preprint arXiv:2410.04715.
[3]Yang, Y., Wang, H., Wen, M., Mo, X., Peng, Q., Wang, J., & Zhang, W. (2024). P3: A Policy-Driven, Pace-Adaptive, and Diversity-Promoted Framework for data pruning in LLM Training. arXiv preprint arXiv:2408.05541.

**Experimental Designs Or Analyses:**

The experiments are comprehensive. However, there are some critical issues:
1.	The paper only applies the method to the instructions of the SFT data, which I think is a critical issue. In many cases, the responses of the SFT data are even more important.
2.	The training data might be too small to show the significance of the performance scores in the experiments.
3.	There are more related baselines that are not considered, see “Essential References Not Discussed” below.

**Methods And Evaluation Criteria:**

I have the following concerns about the methodology:
1.	Maximizing the n-gram indeed ensures some granular lexical diversity of texts, but the general and more importantly semantic or domain diversity of the texts are not considered. Hence the motivation to apply set cover problems on the n-grams is not that robust.
2.	The quality part relies on the previous work SUPERFILTER, leaving the contribution of the paper mostly to the diversity part.
3.	The priority function in equation (4) is a direct integration of quality and diversity score. A balance between these two might be considered.

**Other Comments Or Suggestions:**

N/A

**Other Strengths And Weaknesses:**

The application of the set cover is novel and the experiments of the paper are comprehensive. However, there are some critical weaknesses of the paper, as discussed above.

**Questions For Authors:**

N/A

**Relation To Broader Scientific Literature:**

This paper mostly applies the set cover optimization on the n-grams of texts to select LLM SFT data. The method of using set cover is novel but I think for general improvement of LLM’s performance, this data selection method would have limited contribution.

**Theoretical Claims:**

No theoretical claims are found.

---

> ### Author Rebuttal · Authors · 2025-03-30
>
> We sincerely appreciate your thoughtful review of our paper.
> ## 1. Diversity Approach
> We would like to highlight that **our research demonstrates that lexical diversity through n-grams serves as an effective proxy for semantic diversity.** As shown in Figure 2a and 2c, we demonstrated that the subset selected by GraphFilter exhibits significantly higher lexical diversity compared to all the baselines and is as semantically diverse as those diversity-focused methods (e.g. InsTag). **This suggests that our approach effectively captures semantic diversity through lexical diversity.**
>
> The semantic diversity could be incorporated by extending our priority function to $\phi(u) = quality(u) \times LexicalDiversity(u) \times SemanticDiversity(u)$. For semantic diversity, we could measure the average pairwise cosine distance between embeddings of the candidate example $u$ and all previously selected examples. A larger distance would indicate higher semantic novelty. However, adding semantic distance calculations would introduce significant computational overhead, particularly for large datasets. Due to limited resources and time, we were unable to conduct this experiment in this rebuttal. We will explore this direction in our future revision.
> ## 2. Reliance on SuperFilter
> We would like to respectfully clarify that **GraphFilter is designed to be agnostic to the specific quality metric used, requiring only that the metric can be computed on a per-example basis.** We discovered an inaccuracy in the original Table 4 results. The updated results presented below actually strengthen our claims: GraphFilter maintains strong performance across various quality metrics. These results demonstrate that, while SuperFilter works best in our setting, our approach is not fundamentally dependent on it and can effectively leverage different quality assessment methods. **When new quality metrics are introduced, GraphFilter can be easily adapted to incorporate them, as long as they can be computed on a per-example basis.** Additionally, as suggested by `Reviewer ipfn`, we evaluated GraphFilter with the MTLD score as the diversity metric and the SkyworkRM as the quality metric. The results are presented in the updated Table 4 as well. **We observe that GraphFilter is also compatible with different diversity metrics, further validating the robustness and flexibility of our approach.** We will update Table 4 accordingly in our revision.
> | Quality(u)| Diversity(v) | $\mu_{BENCH}$ | $\mu_{LLM}$ | $\mu_{ALL}$ |
> |--|--|--|--|--|
> |SuperFilter | TF-IDF | 50.55 |42.79|47.97|
> |Perplexity| TF-IDF | 49.21 |40.85|46.43|
> |ArmoRM| TF-IDF | 49.01 |41.85|46.61|
> | DEITA| TF-IDF | 49.11 |41.97|46.73|
> | X| TF-IDF | 48.94 |41.87|46.58|
> |SuperFilter |X | 49.52 |41.28|46.78|
> | X|X | 48.27 |40.28|45.61|
> | SuperFilter | MTLD | 50.15 | 42.42 | 47.47 |
> | SkyworkRM| MTLD | 49.51 | 42.01 | 46.93 |
> ## 3. Priority Function
> Our approach treats quality and diversity as equally important. We acknowledge that introducing an explicit weighting parameter could provide additional flexibility. We will include this discussion and corresponding experiments in our revised manuscript.
> ## 4. Instruction-only Application
> **It is important to note that our approach does not ignore responses entirely.** When computing quality scores, we consider both the instruction and its corresponding response, ensuring that instruction-response pairs are of high quality. Furthermore, **every method has its own optimal way of being applied.** As demonstrated in Table 5 of our submission, **applying GraphFilter to instructions yields superior performance compared to applying it to responses or both.** We will include this discussion in our revised manuscript to clarify our design choices.
> ## 5. Small Dataset
> **We followed well-established practices from previous works, many of which select comparable or even smaller proportions of data.** SuperFiltering [1] selects up to 7800 examples from 52K training examples. AlpaGasus [2] selects 9K examples from 52K training examples. DEITA [3] selects 10K example from 300K training examples. In our work, we select 10K examples from 300K examples. This proportion is consistent with the scale of data selection in related works and allows us to conduct a fair comparison. Furthermore, we conducted extensive experiments across multiple budget sizes, as shown in Figure 3. Our approach consistently outperforms the baseline methods across all budget sizes, demonstrating its robustness and effectiveness. We will include these additional details in the revised manuscript.
>
> References:
>
> 1. Superfiltering: Weak-to-strong data filtering for fast instruction-tuning." 2024.
> 2. Alpagasus: Training a better alpaca with fewer data. 2023.
> 3. What makes good data for alignment? a comprehensive study of automatic data selection in instruction tuning. 2023.
>
> ## 6. Missing Literature
> We will incorporate these references in our revised manuscript.

---

> > ### Comment · Reviewer_ZQbQ · 2025-04-07
> >
> > Thank the authors for the response, which addresses some of my concerns. I have raised my score.

---

### Official Review · Reviewer_6GvX · 2025-03-25

**Overall Recommendation:** 3

**Summary:**

This paper presents GRAPHFILTER, a novel data selection method designed to address the challenge of balancing data quality and diversity in SFT. The core idea is to model the dataset as a bipartite graph where sentences are connected to their constituent n-grams. By using a priority function that multiplicatively combines quality and diversity metrics, GRAPHFILTER iteratively selects sentences. This approach aims to maximize n-gram coverage while taking into account the quality of the data. Extensive experiments on three mainstream models across six benchmarks show that GRAPHFILTER outperforms nine baselines in terms of both model performance and computational efficiency.

**Claims And Evidence:**

This paper made the following main claims, which I think is well supported.
- GRAPHFILTER can achieve a better balance between data quality and diversity compared to existing methods.

- It achieves sota performance on multiple benchmarks

- GRAPHFILTER is computational efficient.

- The combination of n - gramsand the multiplicative priority function are crucial for the success of the method.

- Applying GRAPHFILTER to instructions alone yields the best performance, emphasizing the importance of diverse instructions in SFT.

**Essential References Not Discussed:**

N/A

**Experimental Designs Or Analyses:**

Yes. GRAPHFILTER offers a promising approach to balancing quality and diversity in data selection. The experimental validation is strong, but more details are needed regarding the generalizability of the method, the choice of diversity metrics, and the adaptability to different scenarios.

**Methods And Evaluation Criteria:**

Yes. The methods are novel to me and the evaluation criteria is common used.

**Other Comments Or Suggestions:**

N/A

**Other Strengths And Weaknesses:**

Strengths
1. I think the presented method is simple yet effective. Besides, the computational efficiency is impressive.

Weaknesses
1. There is no mention of error bars (or std) in the tables.

2. The impact of different subset sizes (e.g., 1K vs. 100K) as well as settings of hyperparameters on the balance between quality and diversity needs further exploration.

**Questions For Authors:**

1. Would you consider change the name GRAPHFILTER, to GraphFilter? Writing everything in uppercase looks strange to me.

2. How was the choice of using up to trigrams justified? Would variable or larger n - grams improve the results?

3. What is the sensitivity of the hyperparameters?

**Relation To Broader Scientific Literature:**

N/A

**Theoretical Claims:**

N/A

---

> ### Author Rebuttal · Authors · 2025-03-30
>
> We sincerely appreciate your thoughtful review of our paper on GraphFilter.
> ## 1. Generalizability
> GraphFilter demonstrates strong generalizability through consistent performance across three model backbones and six diverse benchmarks. The Magpie dataset used in our experiments is a general dataset covering a wide range of tasks. We will clarify these points in our future revision.
> ## 2. Diversity Metrics
> **GraphFilter is designed to be agnostic to the specific quality and diversity metrics used.** We have demonstrated that GraphFilter is compatible with a wide range of quality metrics, as shown in Table 4. Please note that we carelessly presented incorrect results in Table 4, which we will correct in our future revision. Furthermore, we conducted additional experiments using MTLD as the diversity metric. During the iterative selection process, we maintained the bipartite graph structure but updated the MTLD scores iteratively as n-gram nodes were removed. As shown in the table below, GraphFilter is flexible enough to accommodate different quality and diversity metrics while maintaining strong performance.
> | Quality(u)| Diversity(v) | $\mu_{BENCH}$ | $\mu_{LLM}$ | $\mu_{ALL}$ |
> |--|--|--|--|--|
> |SuperFilter | TF-IDF | 50.55 |42.79|47.97|
> |Perplexity| TF-IDF | 49.21 |40.85|46.43|
> |ArmoRM| TF-IDF | 49.01 |41.85|46.61|
> | DEITA| TF-IDF | 49.11 |41.97|46.73|
> | X| TF-IDF | 48.94 |41.87|46.58|
> |SuperFilter |X | 49.52 |41.28|46.78|
> | X|X | 48.27 |40.28|45.61|
> | SuperFilter | MTLD | 50.15 | 42.42 | 47.47 |
> | SkyworkRM| MTLD | 49.51 | 42.01 | 46.93 |
> ## 3. Adaptability
> GraphFilter can be easily adapted for domain-specific scenarios. This adaptation can be achieved in two ways: (1) **replacing the default general quality metrics with domain-specific ones**, as discussed in Section 3.3, or (2) **introducing a whitelist for domain-specific terms, allowing n-gram nodes containing these terms to be visited multiple times.** These modifications ensure that domain-specific examples receive higher priority during selection. We will discuss these strategies in our future revision.
> ## 4. Error Bars
> We acknowledge the importance of error bars in our results. Due to limited resources and time, we were unable to run more experiments with different random seeds and include error bars in this rebuttal. We will address this issue in the future revision.
> ## 5. Different Subset Sizes and Hyperparameters
> To evaluate how GraphFilter performs across different subset sizes, we conducted experiments with Llama-3-8B, as shown in Figure 3. Our results demonstrate that the quality-focused data selection method has an advantage when the budget is small, while the diversity-focused method performs better with larger budgets. Our approach consistently outperforms the baseline methods across all budget sizes, demonstrating its robustness and effectiveness.
>
> We assume that both quality and diversity are equally important for data selection, so we did not introduce any hyperparameters into the priority function (Equation 4). We acknowledge that introducing such hyperparameters could further improve the method's adaptability. Due to limited computational resources and time, we were unable to provide the results in this rebuttal. We will include these results in the future version.
> ## 6. Method Name
> We will change the name to "GraphFilter" in the future version to improve readability.
> ## 7. Choice of Trigrams
> **Our choice of trigrams was based on our preliminary study to balance between diversity representation, model performance, and computational efficiency.** As shown in the table below, we conducted experiments with n-gram sizes from 1 to 5 using Llama-3-8B. Our results indicate a significant performance improvement when moving from unigrams (n=1) to trigrams (n=3). However, beyond n=3, we observe diminishing or even negative returns. Furthermore, the number of n-gram nodes increases substantially with n (from 0.1M for unigrams to 7.4M for 5-grams), as well as the runtime (from 2.12 hours to 4.58 hours). We will include this analysis in our future revision.
> | n-gram | # of n-gram nodes | Runtime (hrs) | $\mu_{BENCH}$ | $\mu_{LLM}$ | $\mu_{ALL}$ |
> |--|--|--|--|--|--|
> | 1| 0.1M| 2.12| 49.02 | 41.41 | 46.48 |
> | 2| 1.0M| 2.30| 49.58 | 42.14 | 47.31 |
> | 3| 2.6M| 2.48| 50.55 | 42.79 | 47.97 |
> | 4| 4.8M| 3.38| 50.11 | 42.63 | 47.43 |
> | 5| 7.4M| 4.58| 50.44 | 42.81 | 47.95 |
> ## 8. Hyperparameter Sensitivity
> In GraphFilter, the primary hyperparameter is the n-gram size, which we set to 3. This choice is detailed in Section `7. Choice of Trigrams`. Other hyperparameters, such as learning rate and batch size, are kept consistent across all methods to ensure a fair comparison. Furthermore, we acknowledge the potential benefits of introducing an additional hyperparameter to control the balance between quality and diversity in the priority function (Equation 4). Due to limited resources and time, we will include such an analysis in our future revision.

---

### Decision · Program_Chairs · 2025-05-01

**Decision:**

Accept (poster)

**Comment:**

GRAPHFILTER is a novel data selection method for supervised fine-tuning (SFT) of large language models, designed to balance data quality and diversity. It represents the dataset as a bipartite graph linking sentences to their constituent n-grams. It iteratively selects data points using a priority function that combines quality and diversity metrics, aiming for maximum n-gram coverage with high-quality examples.

Reviewers generally agree on the merits of the proposed method, which introduces a novel and compelling approach to data selection, praised for its simplicity and unique formulation, possibly involving bipartite graphs. This method demonstrates strong empirical performance, achieving effective results across various benchmark datasets. A key advantage is its impressive computational efficiency, fast runtimes compared to alternatives, and lower hardware barriers. The majority of concerns have been addressed through the rebuttal, such that the experimental evaluation lacks rigor, or justifies key decisions like the data budget size, the method's scope and approach, relying on potentially superficial n-gram diversity metrics, and inadequately explores alternative quality assessments. However, we highly recommend that the authors incorporate this feedback and revisions into the next version of the paper.